biomaterials/materials science

activated carbon, plasma, electrochemical property

**Authors for correspondence:**
Xiaoyan Zhou
e-mail: zhouxiaoyan@njfu.edu.cn
Weimin Chen
e-mail: cwmwood@163.com

This article has been edited by the Royal Society of Chemistry, including the commissioning, peer review process and editorial aspects up to the point of acceptance.

# Enhancement of the electrochemical properties of commercial coconut shell-based activated carbon by $H_2O$ dielectric barrier discharge plasma

Xin Wang[1,2], Xiaoyan Zhou[1,2], Weimin Chen[1,2],
Minzhi Chen[1,2] and Chaozheng Liu[1,2]

[1]College of Materials and Engineering, Nanjing Forestry University, No.159 Longpan Road, Nanjing 210037, People's Republic of China
[2]Jiangsu Engineering Research Center of Fast-growing Trees and Agri-fiber Materials, Nanjing 210037, People's Republic of China

XW, 0000-0003-0960-7645; CL, 0000-0003-4168-8289

Commercial coconut shell-based activated carbon (CSAC) has low specific capacitance and specific capacitance retention owing to its undeveloped pore structure and low proportion of heteroatoms. In this study, dielectric barrier discharge plasma was used to enhance the specific capacitance and rate capability of CSAC. $H_2O$ was used as an excited medium to introduce oxygen functional groups. The physico-chemical properties of CSAC and CSAC modified by $H_2O$ plasma (HCSAC) were revealed by automated surface area and pore size analysis, Fourier transform infrared spectroscopy, X-ray photoelectron spectroscopy and Raman spectroscopy. Electrochemical work was applied to investigate the electrochemical properties of CSAC and HCSAC. The results obtained showed that plasma modification improved the specific capacitance of CSAC by 64.8% (current density, 1 A $g^{-1}$; electrolyte, 6 M KOH solution) within 100 s. This result is ascribed to the oxygen functional groups introduced to the surface of CSAC. It can also improve the hydrophilicity and wettability of the carbon surface leading to an increase from 76.7% to 84.6% in specific capacitance retention. Furthermore, $H_2O$ plasma modification can introduce oxygen functional groups without destroying the initial pore structures of CSAC. In summary, we provide a simple, fast, environment-friendly modification method to enhance the electrochemical properties of CSAC.

# 1. Introduction

In recent years, biomass-based activated carbon with a well-developed porosity and high specific surface has been widely applied in electrode materials for the production of supercapacitors [1,2]. Coconut is an important resource in tropical areas and it has been used in commercial production of activated carbon owing to the abundance of materials, low price, stable physical and chemical characteristics and environment-friendliness [3]. It should be noted that coconut shell is considered to be a suitable precursor owing to its availability, low price and ability to make mesoporous structures in carbon materials [4,5]. In fact, commercial coconut shell-based activated carbon (CSAC), as a by-product of agricultural production, after simple chemical activation and large scale production in industrial manufacture, is too coarse to be fully used. Also, CSAC mainly consists of mesopores and a low proportion of micropores, showing low specific capacitance and capacitance retention.

To date, a variety of modifications, including wet or gas oxidation, supercritical fluids, electrochemical oxidation, ion or cluster bombardment and plasma modification and activated carbon modification, have been applied to achieve higher electrochemical capacitance by changing the specific surface, porous structure and the type, content and bonding mode of heteroatoms on the surface of activated carbon [6,7]. In general, the heteroatoms, including sulfur, nitrogen, phosphorus and most frequently oxygen, on the surface of activated carbon, play a significant role in the electrochemical performance of activated carbon [8]. Among them, oxygen functional groups, including hydroxyls, carboxyls, phenols, carbonyls and lactones, covalently attached to the surface, can increase surface polarity and hydrophilicity [9]. It has been reported that activated carbon treated with air or cold oxygen plasma processes can be efficiently used as adsorbent material for removing various pollutants [10]. However, the cold plasma modification requires a highly vacuum environment when applied in activated carbon modification, while dielectric barrier discharge (DBD) can be operated at atmospheric pressure [11]. The cold plasma needs a series of vacuum equipment and maintains a vacuum when in motion, which is expensive and energy consuming. Also, previous studies have found that etching the surface layer of activated carbon using DBD plasma resulted in an increase in the capacitance, which is mostly attributed to the creation of functional groups on the surface [7,11,12]. Xu et al. have performed water vapour plasma modification on wheat straw to improve bonding property. Higher wettability and surface polarity were achieved by introducing massive oxygen-containing groups [13]. In this paper, distilled water was used in DBD plasma as a pollution-free medium to enhance the electrochemical properties of CSAC. Furthermore, it is suitable for practical application in that DBD plasma modification can dramatically reduce the processing time and does not require vacuum equipment.

In the current study, DBD plasma was applied using $H_2O$ as an excitable medium in the fast modification of commercial CSAC, which has never been discussed. A variety of methods were used to reveal the changes in the chemical and physical structures of CSAC after plasma modification. The characterizations were significant in the use of industrial by-product by the addition of value and the application of coconut shell in the supercapacitor electrode material.

# 2. Materials and methods

## 2.1. Materials

The precursor material used in this study is commercial CSAC, supplied by Nanjing Wood Linsen Carbon Company. Acetone, ethanol, hydrochloric acid (HCl), potassium hydroxide (KOH), and carbon black, used as a conductive agent, and polytetrafluoroethylene (PTFE) latex (60 wt.%), used as a binder, were purchased from Nanjing Chemical Reagent Co., Ltd (Jiangsu Province, China). All mentioned agents were used as received. The nickel foam (Nanjing Chemical Reagent Co., Ltd) was first immersed in acetone solution with an ultrasonic bath for 1 h then dipped in HCl (5%) for 1 h after ethanol cleaning. Finally, it was cleaned in ethanol again and dried in a vacuum oven at 40°C for 12 h.

## 2.2. Activated carbon modification

CSAC was rinsed repeatedly to remove the ash content and soaked for 24 h with deionized water. Then CSAC was adequately washed with 1 mol l$^{-1}$ HCl, adjusted to neutrality by deionized water and dried at 110°C for 12 h before use. The diagram of the experimental device for the $H_2O$ plasma modification is shown in figure 1. The DBD plasma reactor (CTP-2000 K, China) had a circular parallel plate with a

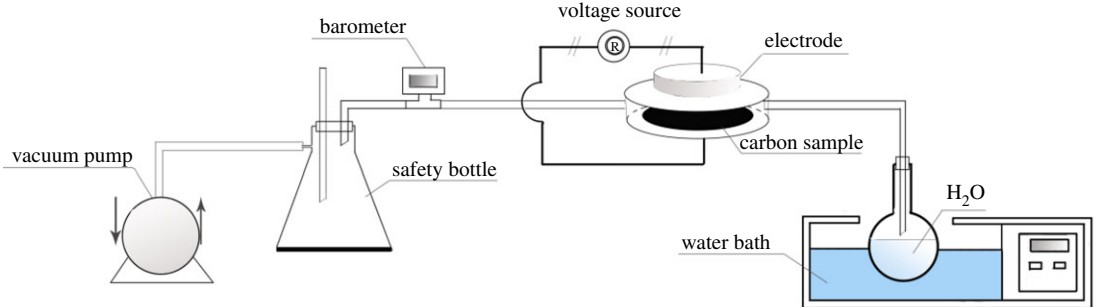

**Figure 1.** Diagram of the experimental device for the $H_2O$ plasma treatment.

diameter of 50 mm and distance between the barrier of 5.0 mm. Approximately 500 mg of activated carbon was placed in the reactor chamber with a thickness of 1 mm each time. A water bath was used to heat $H_2O$ to 70°C and a vacuum pump was applied, keeping the pressure of the reactor chamber at 30 kPa. A number of preliminary experiments were carried out and the experimental conditions (discharge power, 160 W; treating time, 100 s) can achieve better electrochemical performance.

## 2.3. Characterization of activated carbon

Automated surface area and porosity analysis (ASAP 2020, USA) was employed in recording the nitrogen adsorption–desorption isotherms at 77 K. Brunauer–Emmett–Teller (BET) and density functional theory (DFT) were used to reveal the specific surface area and the pore size distribution, respectively [14]. The temperature for degas is 200°C and the time for degas is 15 h. The Fourier transform infrared (FTIR) spectrometer (Nicolet IS10, USA) was used to study the chemical characterization of functional groups using KBr pellets at the scanning range of $4000-400$ cm$^{-1}$. Transmission electron microscopy (TEM, JEOL 2100) was used to observe the morphology of the samples. The carbon samples were added into propanol and sonicated before dispersing on copper grid film before testing. X-ray photoelectron spectroscopy (XPS, AXIS Ultra DLD, Japan) was applied to investigate the surface atomic compositions and chemical functional groups of activated carbon. The C 1 s was deconvoluted into several components using XPSPEAK Software (v. 4.0). The composition of C, N, O elements was determined by the low-resolution spectra (0–1200 eV). The relative proportion of surface oxygen, nitrogen and carbon elements was calculated by the integrated area of each element divided by the integrate area of all the elements (C, O, N), respectively (XPS spectra of CSAC and CSAC modified by $H_2O$ plasma (HCSAC) are available in the electronic supplementary material, figure S1). Raman spectra of activated carbon were recorded by Raman spectroscopy (DXR532, USA) with laser radiation at a wavelength of 532 nm.

## 2.4. Preparation and electrochemical measurements of the electrodes

Activated carbon, carbon black and PTFE were mixed with a weight ratio of 8 : 1 : 1 in an agate mortar to obtain a homogeneous slurry. The slurry was pressed onto a piece of nickel foam with an apparent area of 1 cm$^2$ and dried at 60°C for 12 h. The resulting nickel foam was pressed under a pressure of 15 MPa to assemble the working electrodes. All electrochemical measurements were carried out on an electrochemical workstation (Gamry, USA). For a three-electrode system in 6.0 M KOH aqueous solution electrode, CSAC and HCSAC were used as the working electrodes, and the Hg/HgO electrode and the platinum foil were selected as the reference and counter electrode, respectively. Cyclic voltammetry (CV) and galvanostatic charge–discharge (GCD) measurements were performed over a potential range from −1 to 0 V. The electrochemical impedance spectroscopy (EIS) measurements were recorded in a frequency range from 100 kHz to 10 mHz with an amplitude of 5 mV at open circuit potential. The specific capacitance of the electrodes ($C$) was calculated by the following equation:

$$C = \frac{I \times \Delta t}{\Delta V \times m},$$ (2.1)

where $C$ is the specific capacitance (F g$^{-1}$), $I$ is the constant charge–discharge current (A), $t$ is the discharge time (s), $V$ is the total change in voltage (V), and $m$ is the mass of the active material in an electrode (g).

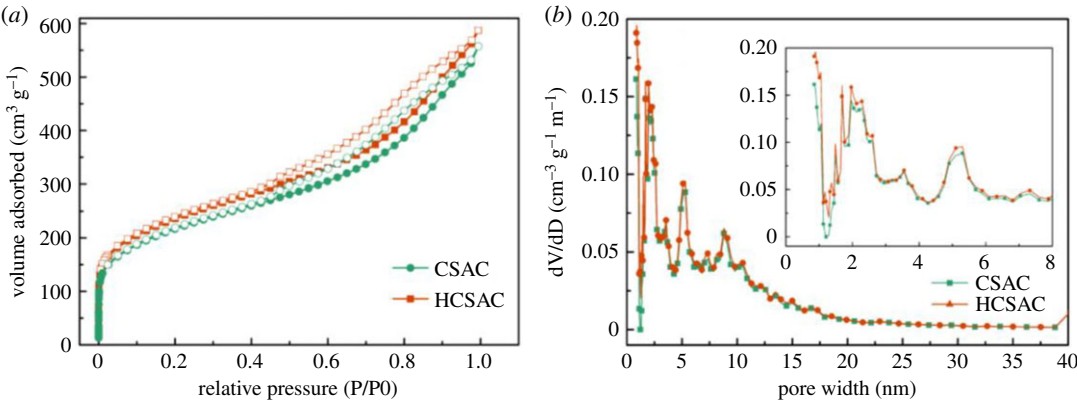

**Figure 2.** Pore structure analysis of (*a*) N$_2$ adsorption – desorption isotherms and (*b*) pore size distribution by DFT method.

**Table 1.** Porous structure parameters of the activated carbon samples.

| sample | BET surface area (m$^2$ g$^{-1}$) | total pore volume (cm$^3$ g$^{-1}$) | micropore volume (cm$^3$ g$^{-1}$) | mesopore volume (cm$^3$ g$^{-1}$) | average pore size (nm) |
|---|---|---|---|---|---|
| CSAC | 779.8 | 0.862 | 0.134 | 0.728 | 4.422 |
| HCSAC | 846.0 | 0.908 | 0.146 | 0.762 | 4.295 |

# 3. Results and discussion

## 3.1. Porous structure characterization

Figure 2 shows N$_2$ adsorption–desorption isotherms and the DFT pore size distributions of CSAC and HCSAC. Both the CSAC and HCSAC exhibit similar combined type I and IV isotherms with small H4 type hysteresis according to the IUPAC classification [15,16]. An almost vertical adsorption line at the relative pressure below 0.10 and the appearance of a hysteresis loop at the pressure range of 0.40–0.99 indicated the existence of micropores and mesopores, respectively [17]. Furthermore, a slight increment in adsorption quantity at the relative pressure near 1 proved that a limited quantity of macropores are induced in the samples [18]. The coexistence of micropores, mesopores and macropores has a positive effect on the electrochemical performance of activated carbon because the interconnected pore structures provide the electrolyte ion with diffusion paths along different directions [19]. It can be seen from table 1 that CSAC and HCSAC that mainly consist of mesopores, and rapid DBD plasma etching process, lead to slight improvement on the BET surface area of the samples (from 779.8 to 846 m$^2$ g$^{-1}$). The size of micropores for the samples displays a multimodal distribution with three peaks at 1.5, 1.7 and 2 nm, and the ratio of micropores has a slight increment after plasma modification, which leads to the increase in BET surface area. It is observed that DBD plasma modification hardly damages the hierarchically pore structures to CSAC.

## 3.2. Morphology and amorphous structure

Figure 3 demonstrates two obvious bonds corresponding to the G-band (1530–1610 cm$^{-1}$) and D-band (1320–1370 cm$^{-1}$) of CSAC and HCSAC. Based on previous studies, the G-band represents graphite in-plane vibrations, while the D-band is attributed to the breathing vibrations of sp$^2$ rings or the double-resonance Raman process in disordered carbon structure [20–22]. The broad D-band indicates that both the samples of activated carbon exhibit disordered domains and poor graphitization [23]. The relative intensity ratio of the D-band to G-band ($I_D/I_G$) has been calculated because the $I_D/I_G$ represents the graphitic degree of the carbon materials. It can be observed that the $I_D/I_G$ value increases from 0.95 to 0.98; the slight increment means DBD plasma modification hardly destroys the graphitic order. The morphology of CSAC and HCSAC was further studied by TEM, as shown in

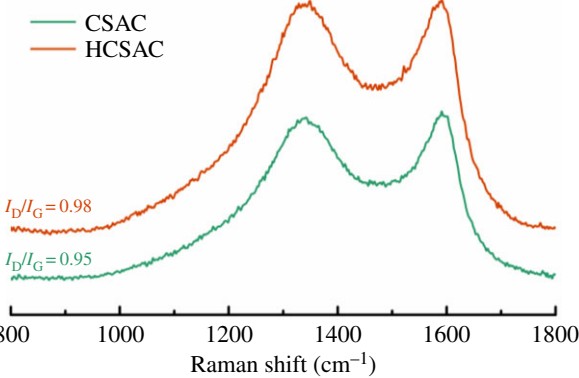

**Figure 3.** Raman spectra of CSAC and HCSAC.

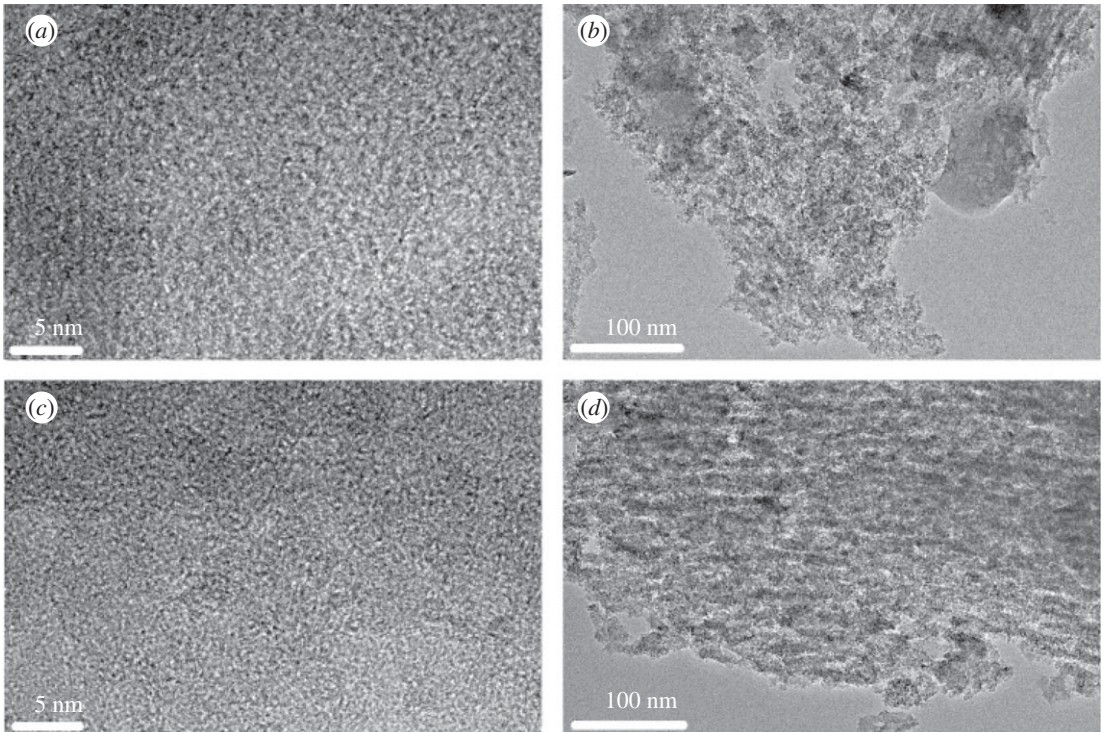

**Figure 4.** TEM images at different magnifications (a,b) of CSAC and (c,d) of HCSAC.

figure 4. Massive nanopores were observed and distributed randomly in carbon samples. No obvious difference was observed between CSAC and HCSAC, which agrees with the BET results.

## 3.3. Chemical characterization

FTIR was used in this study (shown in figure 5) to investigate the surface chemistry properties of CSAC and HCSAC. The obvious peak at $3417 \, \mathrm{cm^{-1}}$ is associated with the −OH stretching vibration due to the surface adsorbed moisture [24]. The peak located at $2925 \, \mathrm{cm^{-1}}$ is attributed to $CH_3$ bands which are consistent with the C−H stretching appearing at $1382 \, \mathrm{cm^{-1}}$ in fingerprints. Another peak located at $1575 \, \mathrm{cm^{-1}}$ reveals the presence of C = C aromatic bonds among samples [25]. The band that appeared at $1022 \, \mathrm{cm^{-1}}$ is designed to be the C−O bond from esters and ethers [24]. The enhanced intensity of the vibration peaks located at $2925 \, \mathrm{cm^{-1}}$ indicates that alkyl groups are included in samples after plasma modification [18].

The surface elemental compositions of CSAC and HCSAC were further investigated by XPS, as shown in figure 6. As shown in figure 6a,b, the C 1 s spectrum can be resolved into four individual peaks based on the previous literature (284.8 eV, $sp^2$ carbon bonds of −C−C−; 285.9 eV, $sp^3$ carbon

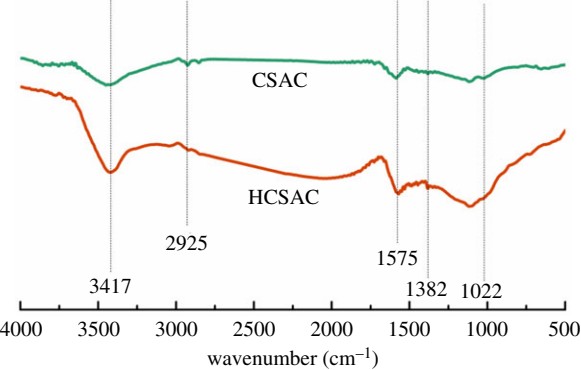

**Figure 5.** FTIR spectra of CSAC and HCSAC.

bonds of −C−O−; 287.4 eV, carbonyl peak (−C = O); 289.1 eV, carboxyl peak) [26–28]. As shown in figure 6c,d, the O 1 s spectrum of CSAC and HCSAC can be resolved into three individual peaks based on the previous literature (531.3 eV, carbonyl and/or quinone; 532.7 eV, hydroxyl and/or ether; 534.0 eV, chemisorbed oxygen and/or water) [29]. As shown in figure 6e,f, the N 1 s spectrum can be fitted by three peaks (398.2 eV, pyridinic nitrogen; 399.9 eV, pyrrolic nitrogen; 400.9 eV, quaternary nitrogen) [30]. The nitrogen content of HCSAC shows slight changes in comparison with CSAC. Table 2 presents the composition of the surface atomic elements and the chemical functional groups of CSAC and HCSAC. A 60.4% increment is observed in the content of O after DBD plasma modification, indicating the presence of more oxygen functional groups on the HCSAC surface. This result may be ascribed to hydroxyl and oxygen ions excited from the $H_2O$ molecule. It has been reported that the increase in the content of carbonyl (−C = O) is beneficial to the electrochemical properties of activated carbon [31]. Furthermore, the obvious increment of oxygen functional groups was considered to facilitate electrochemical redox activity and improve the hydrophilicity and wettability of the material surface, which contribute to the improvement in pseudo-capacitance and electrical double layer capacitance (EDLC) [32]. The three-dimensional schematic model of oxygen functional groups and nitrogen functional groups of HCSAC are presented in figure 7.

## 3.4. Electrochemical properties

The electrochemical properties of CSAC and HCSAC electrodes are studied by CV measurement (electrolyte, 6.0 M KOH aqueous; potential window, −1.0 to 0 V; testing system, three-electrode system). 6.0 M KOH aqueous electrolyte is selected due to its high ionic conductivities [33]. It can be concluded that the $H_2O$ plasma has the ability to improve the electrochemical performance of CSAC within a short time (available in the electronic supplementary material, figure S2a−f). Figure 8a shows the CV plots of the CSAC and HCSAC electrodes at a sweep rate of 10 mV s$^{-1}$. The CV plots of the foam nickel (current collector) are known beforehand, which produces less capacitance than other samples. As for the carbon electrodes, all the CV diagrams display a quasi-rectangular shape, which could be mainly ascribed to EDLC and supplemented by pseudo-capacitance owing to oxygen functional groups on the carbon surface. The additional pseudo-capacitance is attributed to oxygen functional groups. Previous studies demonstrate that some oxygen functional groups can directly participate in faradaic reactions not only in acidic medium, but also in alkaline medium [34,35]. The increased oxygen functional groups play an important role in the enhancement of capacitance via reversible redox actions in alkaline medium. The inductive effects of bond structures of oxygen-containing functional groups are capable of causing the redistribution of electrons and polarization of some bonds. The electric potential-induced redox reactions of polarized sites proceed through the simultaneous reversible gaining/losing of electrons and adsorption/desorption of protons, respectively [36,37]. In comparison with that in CSAC electrodes, the increase in the redox current in the CV curve at around −0.6 V was mainly contributed by faradaic pseudo-capacitance due to the oxygen functional groups introduced by plasma modification [38,39]. The presence of surface oxygen functional groups provided strong polar sites that would adsorb water molecules and thus hinder the migration of electrolytes in pores [34]. Generally, HCSAC demonstrated an obvious increase in both EDLC and pseudo-capacitance in CV curves of which the integrated area is used for calculating specific electrochemical capacitance. It has been proved that the increased oxygen-containing groups can

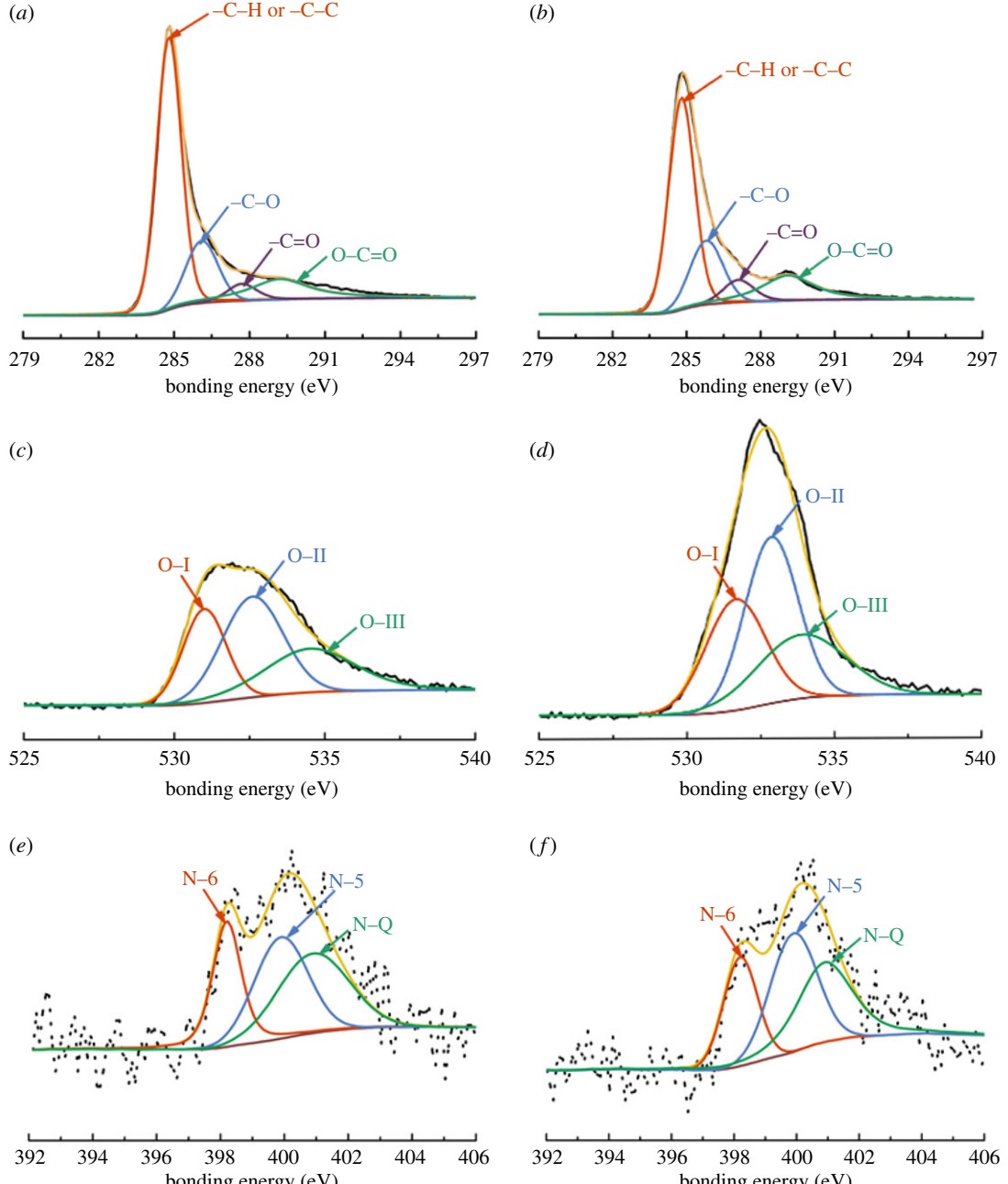

**Figure 6.** (a,b) High-resolution XPS of C 1 s peaks of CSCA and HCSAC. (c,d) High-resolution XPS of O 1 s peaks of CSCA and HCSAC. (e,f) High-resolution XPS of N 1 s peaks of CSAC and HCSAC.

improve the surface wettability by the electrolyte and maximize the electroactive surface area [38–40]. The GCD curves (figure 8b) of CSAC and HCSAC at a current density of 1 A g$^{-1}$ show typical triangular shapes. Owing to the oxygen functional groups in HSCAC, the GCD curves are imperfectly symmetrical but slightly distorted, which was consistent with the CV graphs. A high specific capacitance of 155.2 F g$^{-1}$ higher than CSAC by 64.8% is attained at the current of 1 A g$^{-1}$. To further comprehend the capacitive behaviour of SCAC and HSCAC, EIS was performed over a frequency range from 10 kHz to 10 mHz (figure 8c). The Nyquist plots of CSAC and HSCAC electrodes exhibit similar shapes, in which short x-intercept, small diameter of an inconspicuous semicircle and high slope of the straight line are indicative of an ideal capacitive performance. In the low-frequency region, the line with a slope of 45° accorded with the Warburg resistance of electrolyte ions to the electrode surface. The more vertical and short line of HCSAC over CSAC electrode illustrates electrolyte ions transferring to the pore of activated carbon through a short diffusion path because of higher hydrophilicity and developed pore structures after modification. The intercept on the real axis is related to the internal

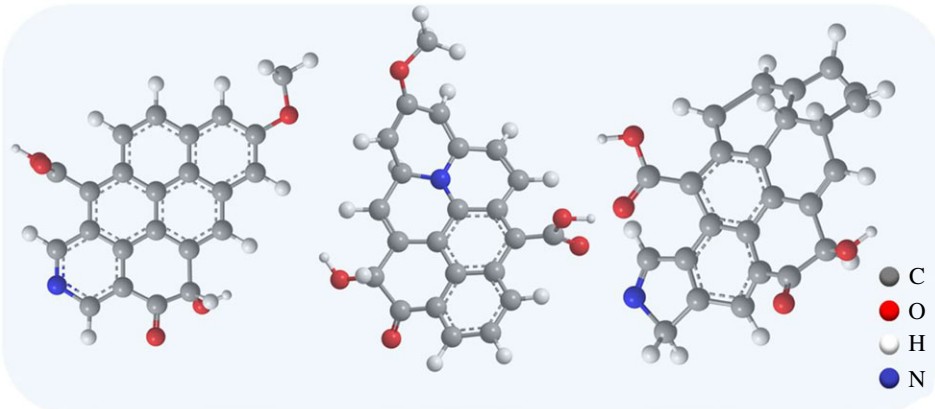

**Figure 7.** Three-dimensional schematic model of the functional groups of HCSAC.

**Table 2.** Composition of surface atomic elements and chemical groups of CSAC and HCSAC.

| | assignment | CSAC | HCSAC |
|---|---|---|---|
| **components (%)** | | | |
| *C 1 s* | | | |
| C 1 | −C−C− or −C−H | 61.5 | 55.1 |
| C 2 | −C−O | 17.7 | 18.2 |
| C 3 | −C＝O | 5.1 | 8.9 |
| C 4 | O−C＝O | 15.7 | 17.8 |
| *O 1 s* | | | |
| O 1 | carbonyl (−C＝O) and/or quinone (marked as O−I) | 27.6 | 32.5 |
| O 2 | hydroxyl (C−OH) and/or ether (C−O−C) (marked as O−II) | 42.0 | 32.3 |
| O 3 | chemisorbed oxygen (COOH) and/or water (marked as O−III) | 30.4 | 35.2 |
| *N 1 s* | | | |
| N 1 | pyridinic nitrogen (marked as N-6) | 25.6 | 27.6 |
| N 2 | pyrrolic nitrogen (marked as N-5) | 38.0 | 35.7 |
| N 3 | quaternary nitrogen (marked as N-Q) | 36.4 | 36.7 |
| **atomic composition (%)** | | | |
| C | | 87.8 | 80.2 |
| O | | 10.6 | 17.0 |
| N | | 1.6 | 2.8 |

resistance ($R_i$), including the resistance of bulk electrolyte, intrinsic active material resistance and their contact resistance with the foam nickel [41]. The $R_i$ of samples was below 0.4 Ω, demonstrating good electrical conductivity of materials. The increased internal resistance of the HCSAC electrode can be ascribed to the presence of surface oxides and thus increase the ohmic resistance along the axial direction of micropores, which show more quantity in HCSAC than in CSAC [34]. With the frequency decreasing, the semicircle represents the charge transfer resistance ($R_{ct}$) between the interface of electrolyte and electrode. The smaller diameter in the semicircle of HCSAC electrodes illustrated the lower $R_{ct}$ at HCSAC−electrolyte interfaces mainly due to the higher hydrophilicity after the introduction of oxygen functional groups that enabled rapid ion diffusion and ideal capacitive behaviours. As shown in figure 8$d$, an increase from 76.7% to 84.6% in capacitance retention was observed when the current density increased from 0.5 to 10 A g$^{-1}$ after plasma modification. This result is attributed to a higher microporous utilization rate at high current density after plasma modification. The life cycles of HCSAC at different current densities for 1000 cycles to prove the reusability of HCSAC can be seen in figure 9. The capacitance retention of HSCAC electrodes maintains over 97% in the 6.0 M

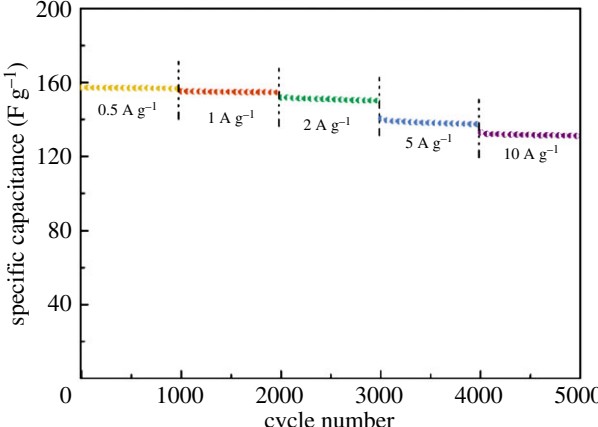

**Figure 8.** (*a*) Cyclic voltammetry curves of CSAC, HCSAC and nickel foam in a three-electrode system with 6 mol l$^{-1}$ KOH aqueous solution electrolyte from $-1.0$ to 0 V at 10 mV s$^{-1}$; (*b*) galvanostatic charge–discharge curves of CSAC and HCSAC at 1 A g$^{-1}$; (*c*) Nyquist plots of CSAC and HCSAC. The inset is the detail with enlarged scale; and (*d*) dependence of specific capacitance of CSAC and HCSAC on current density from 0.5 to 10 A g$^{-1}$.

**Figure 9.** Life cycles of HCSAC electrode at different current densities for 1000 cycles.

KOH electrolyte. The high capacitance retention indicates an ideal interconnected porous structure generating high ion transmission efficiency in HCSAC.

## 4. Conclusion

A simple, fast and environment-friendly method was provided to enhance the specific capacitance and capacitance retention of CSAC. The BET surface area (846.0 m$^2$ g$^{-1}$) of HCSAC shows an unremarkable change in comparison with the initial material. A 60.4% higher surface oxygen content is observed compared with the untreated materials, indicating massive free radicals are generated and introduced

to the carbon surface during $H_2O$ plasma modification. Quinolyl ($-C = O$) exhibits a 74.5% increment after modification, which is capable of facilitating electrochemical redox activity and improving the hydrophilicity and wettability of the material surface. $H_2O$ plasma modification improves the specific capacitance of CSAC by 64.8% at a current density of $1 A g^{-1}$. A higher specific capacitance retention of 84.6% is also achieved. Massive free radicals were generated and introduced to the carbon surface during $H_2O$ plasma modification; in particular, quinonyl ($-C = O$) was successfully incorporated after plasma modification. Furthermore, $H_2O$ plasma modification can introduce oxygen functional groups without destroying the initial porous structure of CSAC.

Data accessibility. Our data are deposited at Dryad: http://dx.doi.org/10.5061/dryad.9mt7743 [42]. Electronic supplementary material is available online at: https://figshare.com/s/7bd218d714eac3913b17.

Authors' contributions. X.W. and X.Z. carried out the molecular laboratory work, participated in data analysis, carried out sequence alignments, participated in the design of the study and drafted the manuscript; M.C. carried out the statistical analyses; C.L. collected field data; X.W. and W.C. conceived of the study, designed the study, coordinated the study and helped draft the manuscript. All authors gave final approval for publication.

Competing interests. We declare we have no competing interests.

Funding. Financial support came from the Natural Science Foundation of Jiangsu Province (grant no. BK20161524), the Program for 333 Talents Project in Jiangsu Province (grant no. BRA2016381), the National Natural Science Foundation of China (grant no. 31400515) and the Priority Academic Program Development of Jiangsu Higher Education Institutions (PAPD). Also, this paper was sponsored by the Qing Lan Project.

Acknowledgements. The authors gratefully acknowledge the support from the Research Center for Fast-growing Trees and Agri-fiber Materials of Jiangsu Province in China.

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
