## [Reviewer comments · Royal Society Open Science]

Review History

RSOS-180872.R0 (Original submission)

Review form: Reviewer 1

Is the manuscript scientifically sound in its present form?

Yes

Are the interpretations and conclusions justified by the results?

No

Is the language acceptable?

Yes

Is it clear how to access all supporting data?

Yes

Do you have any ethical concerns with this paper?

No

Have you any concerns about statistical analyses in this paper?

No

Recommendation?

Major revision is needed (please make suggestions in comments)

Comments to the Author(s)

The paper of Wang et al. deals with the Enhancement on the electrochemical properties of commercial coconut shell-based activated carbon by H₂O dielectric barrier discharge plasma. One may deplore that, finally, only one single material has been prepared and investigated. There is neither repetition of the same, in order to check the repeatability, nor synthesis of a series of materials in which one parameter would have been varied with some impact on the final properties. As a result, this is a quite short studies with a very limited content for a full-length paper. Detailed comments follow:

1. Page1, line 32; line 52, 55, the states are contradict each other;
 2. Page 2, section 3.3, This section must be considerably improved because of the dramatic mack of details. Examples are: " the temprature for degass"
 3. section4.1, line 31, the isotherm type, please make sure.
 4. section 4.2, the Figure 2 should be Figure 3. TEM should be Figure 4.
 5. The BET method is known to overestimate the surface area of microporous materials. If the pore size distribution has been obtained by DFT method, why not having applied the same for determining the surface area? The corresponding result should be different and more reliable. The same also applies to the micropore volume, VDR being always overestimated.
 6. Authors should provide the details of determination of O and N content by XPS.
 7. Authors should rewrite the conclusions showing parameters of the final material, which are more important.
 8. How much of weight loss during plasma treatment.
 9. this paper paye close attention on the plasma midification, so only the C1s deconvoluted in Figure 6 is not enough, the N1 and O1 should also be done as well. one more thing, XPS analysis should have allowed a quantitative assignment of each kind of nitrogen/carbon/oxygen instead of just a qualitative approach.
 10. Did the authors study the reusability for electrochemical properties? It is important to cost down the process.
 11. It is necessary to compare the results with some recent literatures.
 12. Deeper discussion of the results should be provided.
- I would encourage the authors to reconsider all of these points and, after a careful review I would encourage a resubmission.

Review form: Reviewer 2

Is the manuscript scientifically sound in its present form?

Yes

Are the interpretations and conclusions justified by the results?

Yes

Is the language acceptable?

Yes

Is it clear how to access all supporting data?

Yes

Do you have any ethical concerns with this paper?

No

Have you any concerns about statistical analyses in this paper?

No

Recommendation?

Major revision is needed (please make suggestions in comments)

Comments to the Author(s)

Comments:

In this work, carbon treated by water vapor plasma is reported for EDLCs electrodes application. The treated carbon based electrodes present a much better capacitance performance than that of untreated. The method is creative, however, the explanation of plasma working mechanism is not clear enough, either the enhancement of capacitance. Upon reviewing, major revision is suggested. A few comments are as following:

Typos like missing space between value and unit (70°C, line 46, P2, 5.0mm, line 47, P2 etc.), capital letter (Rct, line 46, P4) are detected. Check carefully!

Abbreviations should be denoted at their first appearance in main text, e.g. CSAC. EDLC is commonly the abbreviation of “electrical double layer capacitor” rather than electrical double layer capacitance.

The Fig. numbers are in mess, do not match with the description in text.

Fig. 7d (Fig. 6 in manuscript) is not mentioned. Looks like it is the capacitance performance test at different current density, however, more cycles, e.g. 1000 cycles for each current density, should be provided. Otherwise, it is not sufficient to prove the good performance at different current densities.

In electrochemical properties section, the explanation of improved capacitance performance is concerned. It is claimed by authors that the improvement is attributed to pseudo-capacitance owing of oxygen functional groups on carbon surface (line 22-23, P4) besides double layer capacitance. How did the pseudo capacitance of oxygen functional groups work? What is the the possible working mechanism? And any supporting references?

Contact angel should be measured to support the claimed improved hydrophilicity and wettability after plasma treatment;

How is the plasma working condition come up, e.g. 160 W, 100s, 30KPa? How did these factors affect the treatment and consequently to capacitance performance? In my experience, the treatment time is highly related to the structure evolution.

In the XPS analysis result (Table 2), why is the atomic composition of N increasing?

Decision letter (RSOS-180872.R0)

13-Nov-2018

Dear Dr Wang:

Title: Enhancement on the electrochemical properties of commercial coconut shell-based activated carbon by H₂O dielectric barrier discharge plasma

Manuscript ID: RSOS-180872

The editor assigned to your manuscript has now received comments from reviewers. We would like you to revise your paper in accordance with the referee and Subject Editor suggestions which can be found below (not including confidential reports to the Editor). Please note this decision does not guarantee eventual acceptance.

Please submit your revised paper before 06-Dec-2018. Please note that the revision deadline will expire at 00.00am on this date. If we do not hear from you within this time then it will be assumed that the paper has been withdrawn. In exceptional circumstances, extensions may be possible if agreed with the Editorial Office in advance. We do not allow multiple rounds of revision so we urge you to make every effort to fully address all of the comments at this stage. If deemed necessary by the Editors, your manuscript will be sent back to one or more of the original reviewers for assessment. If the original reviewers are not available we may invite new reviewers.

Please also include the following statements alongside the other end statements. As we cannot publish your manuscript without these end statements included, if you feel that a given heading is not relevant to your paper, please nevertheless include the heading and explicitly state that it is not relevant to your work.

- Ethics statement

Please clarify whether you received ethical approval from a local ethics committee to carry out your study. If so please include details of this, including the name of the committee that gave consent in a Research Ethics section after your main text. Please also clarify whether you received informed consent for the participants to participate in the study and state this in your Research Ethics section.

OR

Please clarify whether you obtained the necessary licences and approvals from your institutional animal ethics committee before conducting your research. Please provide details of these licences and approvals in an Animal Ethics section after your main text.

OR

Please clarify whether you obtained the appropriate permissions and licences to conduct the fieldwork detailed in your study. Please provide details of these in your methods section.

On behalf of the Subject Editor Professor Anthony Stace and the Associate Editor Professor Claire Carmalt.

RSC Associate Editor:
Comments to the Author:
(There are no comments.)

RSC Subject Editor:
Comments to the Author:
(There are no comments.)

Reviewers' Comments to Author:
Reviewer: 1

Comments to the Author(s)

The paper of Wang et al. deals with the Enhancement on the electrochemical properties of commercial coconut shell-based activated carbon by H₂O dielectric barrier discharge plasma. One may deplore that, finally, only one single material has been prepared and investigated. There is neither repetition of the same, in order to check the repeatability, nor synthesis of a series of materials in which one parameter would have been varied with some impact on the final properties. As a result, this is a quite short studies with a very limited content for a full-length paper. Detailed comments follow:

1. Page1, line 32; line 52, 55, the states are contradict each other;
2. Page 2, section 3.3, This section must be considerably improved because of the dramatic lack of details. Examples are: " the temprature for degass"
3. section4.1, line 31, the isotherm type, please make sure.
4. section 4.2, the Figure 2 should be Figure 3. TEM should be Figure 4.
5. The BET method is known to overestimate the surface area of microporous materials. If the pore size distribution has been obtained by DFT method, why not having applied the same for determining the surface area? The corresponding result should be different and more reliable. The same also applies to the micropore volume, VDR being always overestimated.
6. Authors should provide the details of determination of O and N content by XPS.
7. Authors should rewrite the conclusions showing parameters of the final material, which are more important.

8. How much of weight loss during plasma treatment.
9. this paper paye close attention on the plasma midification, so only the C1s deconvoluted in Figure 6 is not enough, the N1 and O1 should also be done as well.
one more thing, XPS analysis should have allowed a quantitative assignment of each kind of nitrogen/ carbon/oxygen instead of just a qualitative approach.
10. Did the authors study the reusability for electrochemical properties? It is important to cost down the process.
11. It is necessary to compare the results with some recent literatures.
12. Deeper discussion of the results should be provided.

I would encourage the authors to reconsider all of these points and, after a careful review I would encourage a resubmission.

Reviewer: 2

Comments to the Author(s)

Comments:

In this work, carbon treated by water vapor plasma is reported for EDLCs electrodes application. The treated carbon based electrodes present a much better capacitance performance than that of untreated. The method is creative, however, the explanation of plasma working mechanism is not clear enough, either the enhancement of capacitance. Upon reviewing, major revision is suggested. A few comments are as following:

Typos like missing space between value and unit (70°C, line 46, P2, 5.0mm, line 47, P2 etc.), capital letter (Rct, line 46, P4) are detected. Check carefully!

Abbreviations should be denoted at their first appearance in main text, e.g. CSAC. EDLC is commonly the abbreviation of “electrical double layer capacitor” rather than electrical double layer capacitance.

The Fig. numbers are in mess, do not match with the description in text.

Fig. 7d (Fig. 6 in manuscript) is not mentioned. Looks like it is the capacitance performance test at different current density, however, more cycles, e.g. 1000 cycles for each current density, should be provided. Otherwise, it is not sufficient to prove the good performance at different current densities.

In electrochemical properties section, the explanation of improved capacitance performance is concerned. It is claimed by authors that the improvement is attributed to pseudo-capacitance owing of oxygen functional groups on carbon surface (line 22-23, P4) besides double layer capacitance. How did the pseudo capacitance of oxygen functional groups work? What is the the possible working mechanism? And any supporting references?

Contact angel should be measured to support the claimed improved hydrophilicity and wettability after plasma treatment;

How is the plasma working condition come up, e.g. 160 W, 100s, 30KPa? How did these factors affect the treatment and consequently to capacitance performance? In my experience, the treatment time is highly related to the structure evolution.

In the XPS analysis result (Table 2), why is the atomic composition of N increasing?

Author's Response to Decision Letter for (RSOS-180872.R0)

See Appendix A.

RSOS-180872.R1 (Revision)

Review form: Reviewer 1

Is the manuscript scientifically sound in its present form?

Yes

Are the interpretations and conclusions justified by the results?

Yes

Is the language acceptable?

Yes

Is it clear how to access all supporting data?

Not Applicable

Do you have any ethical concerns with this paper?

No

Have you any concerns about statistical analyses in this paper?

No

Recommendation?

Accept as is

Comments to the Author(s)

none

Review form: Reviewer 2

Is the manuscript scientifically sound in its present form?

Yes

Are the interpretations and conclusions justified by the results?

Yes

Is the language acceptable?

Yes

Is it clear how to access all supporting data?

Yes

Do you have any ethical concerns with this paper?

No

Have you any concerns about statistical analyses in this paper?

No

Recommendation?

Accept with minor revision (please list in comments)

Comments to the Author(s)

I am satisfied with the modifications made by authors according to my comments, and it can be accepted on condition the supplementary figures, discussions and references are wrapped up with the manuscript. Otherwise, the manuscript is quite short, hardly to be a full-length research paper with few substantial contents. Besides, the response to comment 9 of reviewer #1 is concerned. Did the authors measure the mass of samples before and after plasma treatment? Besides, 160 W is not low and there will be mass loss based on my experience on plasma processing.

Decision letter (RSOS-180872.R1)

03-Jan-2019

Dear Dr Wang:

Title: Enhancement on the electrochemical properties of commercial coconut shell-based activated carbon by H₂O dielectric barrier discharge plasma
Manuscript ID: RSOS-180872.R1

Thank you for submitting the above manuscript to Royal Society Open Science. On behalf of the Editors and the Royal Society of Chemistry, I am pleased to inform you that your manuscript will be accepted for publication in Royal Society Open Science subject to minor revision in accordance with the referee suggestions. Please find the reviewers' comments at the end of this email.

The reviewers and handling editors have recommended publication, but also suggest some minor revisions to your manuscript. Therefore, I invite you to respond to the comments and revise your manuscript.

Please also include the following statements alongside the other end statements. As we cannot publish your manuscript without these end statements included, if you feel that a given heading is not relevant to your paper, please nevertheless include the heading and explicitly state that it is not relevant to your work. We have included a screenshot example of the end statements for reference.

- Ethics statement

Please clarify whether you received ethical approval from a local ethics committee to carry out your study. If so please include details of this, including the name of the committee that gave consent in a Research Ethics section after your main text. Please also clarify whether you received informed consent for the participants to participate in the study and state this in your Research Ethics section.

OR

Please clarify whether you obtained the necessary licences and approvals from your institutional animal ethics committee before conducting your research. Please provide details of these licences and approvals in an Animal Ethics section after your main text.

OR

Please clarify whether you obtained the appropriate permissions and licences to conduct the fieldwork detailed in your study. Please provide details of these in your methods section.

Because the schedule for publication is very tight, it is a condition of publication that you submit the revised version of your manuscript before 12-Jan-2019. Please note that the revision deadline will expire at 00.00am on this date. If you do not think you will be able to meet this date please let me know immediately.

Best wishes,

Dr Laura Smith
Publishing Editor, Journals

On behalf of the Subject Editor Professor Anthony Stace and the Associate Editor Professor Claire Carmalt.

RSC Associate Editor:
Comments to the Author:
(There are no comments.)

RSC Subject Editor:
Comments to the Author:
(There are no comments.)

Reviewer comments to Author:
Reviewer: 2

Comments to the Author(s)

I am satisfied with the modifications made by authors according to my comments, and it can be accepted on condition the supplementary figures, discussions and references are wrapped up with the manuscript. Otherwise, the manuscript is quite short, hardly to be a full-length research paper with few substantial contents. Besides, the response to comment 9 of reviewer #1 is concerned. Did the authors measure the mass of samples before and after plasma treatment? Besides, 160 W is not low and there will be mass loss based on my experience on plasma processing.

Reviewer: 1

Comments to the Author(s)
none

Author's Response to Decision Letter for (RSOS-180872.R1)

See Appendix B.

Decision letter (RSOS-180872.R2)

14-Jan-2019

Dear Dr Wang:

Title: Enhancement on the electrochemical properties of commercial coconut shell-based activated carbon by H₂O dielectric barrier discharge plasma
Manuscript ID: RSOS-180872.R2

It is a pleasure to accept your manuscript in its current form for publication in Royal Society Open Science. The chemistry content of Royal Society Open Science is published in collaboration with the Royal Society of Chemistry.

On behalf of the Subject Editor Professor Anthony Stace and the Associate Editor Professor Claire Carmalt.

RSC Associate Editor
Comments to the Author:
(There are no comments.)

Reviewer(s)' Comments to Author:

Appendix A

Responses to reviewer's comments

We appreciate the detailed and helpful comments and suggestions which are very helpful to improve the quality of our manuscript. The manuscript (RSOS-180872) has been carefully revised as required. All new and revised contents are highlighted with yellow in our revised version. We hope that the correction will meet with your approval. The point-by-point answers to the comments and suggestion were listed below.

Reviewer #1

[1] Only one single material has been prepared and investigated. There is neither repetition of the same, in order to check the repeatability, nor synthesis of a series of materials in which one parameter would have been varied with some impact on the final properties. As a result, this is a quite short studies with a very limited content for a full-length paper.

Response: According to your helpful suggestion, we have provided more electrochemical performance data of samples prepared under different time (50 s, 100 s, 150 s, 200 s, 300 s) and power (50 W, 100 W, 130 W, 160 W, 200 W), please see the figure below. We have conducted a series of preliminary experiment on electrochemical performance to select the optimum condition (100 s, 160 W). From the results shown below, it can be concluded that the H₂O dielectric barrier discharge (DBD) plasma has ability to improve electrochemical performance of commercial coconut shell-based activated carbon (CSAC) in a short time. The exhaustive study is very meaningful that we are going to investigate in the future.

The electrochemical performance of CSAC modified with H₂O plasma under different time and different power is shown below (Fig. 1). The cyclic voltammetry (CV) plots at the scan rate at 10 mV s⁻¹ of CSAC and CSAC modified with H₂O plasma are shown in Fig. 1(a-b). The integrated area of the sample D-100 W-100 s (D means DBD plasma modification, the first number is modification power, the second number is modification time) is larger than other sample in the same power, indicating that 100 s is a more suitable modification time. With modification power changing from 50 W to 200 W, the sample D-160 W-100 s exhibits the largest integrate area among all the samples at the same modification time of 100 s. The largest integrate area of D-160 W-100 s indicates that the sample shows the larger specific capacitance than other samples. Furthermore, the galvanostatic charge-discharge (GCD) curves at the current density of 1 A g⁻¹ of CSAC and CSAC modified with H₂O plasma are shown in Fig. 1(c-d). After calculation by the formula below, we come up the conclusion that D-160 W-100 s exhibits the excellent specific capacitance (specific capacitance of 155.2 F g⁻¹ at the current density of 1 A g⁻¹) among all the samples, which is consistent with CV curves. The Nyquist plots of CSAC and CSAC modified with different power exhibit similar shapes, shown in Fig. 1e. The internal resistance (R_i) of samples were below 0.4 Ω ,

demonstrating a good electrical conductivity of materials. The increased R_i of the sample D-160 W-100 s can be ascribed to the presence of surface oxides and thus increase the ohmic resistance along the axial direction of micropores. Also, the smaller diameter in the semicircle of the sample D-160 W-100 s illustrates that the lower charge transfer resistance (R_{ct}) at electrolyte interfaces, owing to the higher hydrophilicity after oxygen functional groups introduced. Fig. 1f demonstrates the rate capability (capacitance retention from the current density of 0.5 A g^{-1} to 10 A g^{-1}) of CSAC and CSAC modified under different power. The higher rate capability indicates that the sample is capable of maintaining excellent charge and discharge characteristics at high current density. In summary, we perform other detailed characterization of CSAC modified by H_2O plasma under modification time of 100 s and power of 160 W (HCSAC) based on the electrochemical performance. We keep the pressure of the reactor chamber to 30 KPa, mainly aiming to pump water into the reactor chamber instead of maintaining the vacuum environment. Based on the above results, we select the conditions (discharge power, 160 W; treating time, 100 s; the pressure, 30 KPa) to make detailed analysis. We supply the result in supporting information.

The specific capacitance of the electrodes (C), was calculated by the following equation:

$$C = \frac{I \times \Delta t}{\Delta V \times m}$$

Where C is specific capacitance (F g^{-1}), I is the constant charge-discharge current (A), Δt is the discharge time (s), ΔV is the total change in voltage (V), m is the mass of the active material in an electrode (g)

Fig. 1 (a-b) CV curves of CSAC and CSAC modified with DBD H₂O plasma in a three-system with 6 M KOH aqueous electrolyte at the scan rate of 10 mV s⁻¹. (c-d) GCD curves of CSAC and CSAC modified with DBD H₂O plasma in a three-electrode system with 6 M KOH aqueous electrolyte at the current density of 1 A g⁻¹. (e) Nyquist plots of CSAC and CSAC modified at different power electrodes. The inset is the detail with enlarged scale (f) Rate capability of CSAC and CSAC modified at different power at the current density from 0.5 to 10 A g⁻¹.

[2] Page1, line 32; line 52, 55, the states are contradict each other.

Response: Based on your significant suggestion, I realize that the ‘Introduction’ section is incorrectly in expression and revised below. The sentence of ‘It should be noted that activated carbon synthesized from coconut shell is considered better because of its mesoporous structure which makes it suitable for its application in supercapacitor as electrode materials’ has been revised into ‘It should be noted that coconut shell is considered to be a suitable precursor owing to its availability, low price and ability to

create mesoporous structure in carbon materials’.

Page1, line 55 --- ‘It should be noted that coconut shell is considered to be a suitable precursor owing to its availability, low price and ability to create mesoporous structure in carbon materials.’

[3] Page 2, section 3.3, This section must be considerably improved because of the dramatic lack of details. Examples are: " the temperature for degass".

Response: According to your helpful suggestion, we have added ‘The temperature for degas is 200 °C and the time for degas is 15 h’ in ‘Materials and Methods’ section. Similar issues have also addressed through our manuscript and the related revised content is present below.

Page 3, line 3 --- ‘The temperature for degas is 200 °C and the time for degas is 15 h.’

Page 3, line 5 --- ‘Transmission electron microscopy (TEM, JEOL 2100) were used to observe the morphology of the samples. The carbon samples were added into propanol and sonicated before dispersing on copper grid film before testing.’

Page 3, line 9 --- ‘The composition of C, N, O elements were determined by the low-resolution spectra (0-1200 eV).’

[4] section4.1, line 31, the isotherm type, please make sure.

Response: After we consider your significant suggestion, we make sure that the isotherm show a combined type I and IV isotherm with small H4 type hysteresis according to the IUPAC classification. We have revised the isotherm type as shown below^[1,2].

Page 3, line 41 --- ‘Both the CSAC and HCSAC exhibit similar combined type I and IV isotherms with small H4 type hysteresis according to the IUPAC classification.’

Reference:

[1] Zhao Y Q, Lu M, Tao P Y, et al. Hierarchically porous and heteroatom doped carbon derived from tobacco rods for supercapacitors[J]. Journal of Power Sources. 2016, 307: 391-400.

[2] Bhattacharjya D, Yu J S. Activated carbon made from cow dung as electrode material for electrochemical double layer capacitor[J]. Journal of Power Sources. 2014, 262(4): 224-231.

[5] section 4.2, the Figure 2 should be Figure 3. TEM should be Figure 4.

Response: According to your helpful suggestions, we carefully checked the valid numerals of our table and figure through the whole paper and corrected them. The related number of figure and table through the whole paper were also revised below.

Page 3, line 41 --- ‘Figure 2 shows N₂ adsorption/desorption isotherms and the DFT pore size distributions of CSAC and HCSAC.’

Page 3, line 57 --- ‘Figure 3 demonstrates two obvious bands corresponding to the G-band (1530-1610 cm^{-1}) and D-band (1320-1370 cm^{-1}) of CSAC and HCSAC.’

Page 4, line 2 --- ‘Morphology of CSAC and HCSAC were further studied by TEM, shown in Fig. 4.’

Page 4, line 7 --- ‘FTIR was used in this study (shown in Fig. 5) to investigate the surface chemistry properties of SCAC and HCSAC.’

Page 4, line 15 --- ‘The surface elemental compositions of CSAC and HCSAC were further investigated by X-ray photoelectron spectroscopy (XPS), shown in Fig. 6.’

Page 4, line 34 --- ‘Fig. 7a shows the CV plots of the CSAC and HCSAC electrodes at a sweep rate of 10mV s^{-1} .’

Page 4, line 51 --- ‘The GCD curves (Fig. 7b) of CSAC and HCSAC at a current density of 1A g^{-1} show typical triangular shapes.’

Page 4, line 56 --- ‘To further comprehend the capacitive behavior of SCAC and HCSAC, EIS test was performed over a frequency range from 10 kHz to 10 mHz (Fig. 7c).’

[6] The BET method is known to overestimate the surface area of microporous materials. If the pore size distribution has been obtained by DFT method, why not having applied the same for determining the surface area? The corresponding result should be different and more reliable. The same also applies to the micropore volume, VDR being always overestimated.

Response: Based on your significant suggestion, we realized that density functional theory (DFT) method is more suitable than Brunauer-Emmett-Teller (BET) method for obtaining the surface area especially for the case that we have already applied DFT method to study the pore size distribution of carbon samples. We provide you with the surface area revealed by DFT method. The DFT surface area of CSAC and HCSAC is 644.0 and 726.8 $\text{m}^2 \text{g}^{-1}$, respectively. (The BET surface area of CSAC and HCSAC is 779.8 and 846 $\text{m}^2 \text{g}^{-1}$, respectively.) We choose BET method instead of DFT method, mainly owing to the fact that BET method is used widely in other reports^[1-3]. It is convenient for us to compare our data with the value in other reports, so that we can understand the activated carbon performance clearly.

Reference:

[1] Zhang J, Jin L, Cheng J, et al. Hierarchical porous carbons prepared from direct coal liquefaction residue and coal for supercapacitor electrodes[J]. Carbon. 2013, 55(2): 221-232.

[2] Wang Q, Yan J, Wang Y, et al. Three-dimensional flower-like and hierarchical porous carbon materials as high-rate performance electrodes for supercapacitors[J]. Carbon. 2014, 67(2): 119-127.

[3] Zhao Y Q, Lu M, Tao P Y, et al. Hierarchically porous and heteroatom doped carbon derived from tobacco rods for supercapacitors[J]. Journal of Power Sources. 2016, 307: 391-400.

[7] Authors should provide the details of determination of O and N content by XPS.

Response: After we considered your helpful suggestions, the details of determination of O and N by X-ray photoelectron spectroscopy (XPS) are provided. The XPS spectra

of CSAC and HCSAC is shown in Fig. 2. As presented in the figure below, the relative proportion of surface oxygen, nitrogen and carbon elements are calculated by the integrate area of each element divided by the integrate area of all the elements (C, O, N), respectively. Also, we have added the details in the ‘Materials and Methods’ section.

Fig. 2 XPS spectra of CSAC and HCSAC

Page 3, line 10 --- ‘The relative proportion of surface oxygen, nitrogen and elements are calculated by the integrate area of each element divided by the integrate area of all the elements (C, O, N), respectively.’

[8] Authors should rewrite the conclusions showing parameters of the final material, which are more important.

Response: Based on your significant suggestion, we have revised the ‘Conclusion’ section and added parameters of the final materials. We add the detailed properties of HCSAC. The related revised content is presented below.

Page 5, line 20 --- ‘The BET surface area ($846.0 \text{ m}^2 \text{ g}^{-1}$) of HCSAC shows unremarkable change in comparison with the initial material.’

Page 5, line 21 --- ‘A 60.4% higher surface oxygen content is observed as compared to the untreated materials, indicating massive free radicals are generated and introduced to the carbon surface during H_2O plasma modification.’

Page 5, line 23 --- ‘Quinolyl ($-\text{C}=\text{O}$) exhibits an 74.5% increment after modification, which is capable of facilitating electrochemical redox activity and improve hydrophilicity and wettability of material surface.’

[9] How much of weight loss during plasma treatment.

Response: There is no weight loss of CSAC during plasma treatment owing to the low discharge power (160W) utilized in our experiment^[1,2]. The BET surface area, total pore volume and average pore size exhibit unremarkable changes after H_2O plasma

modification. The physical structure without significant changes indicates that H₂O plasma treatment can hardly cause weight loss of CASC.

Reference:

[1] Chen M, Zhang R, Tang L, et al. Effect of Plasma Processing Rate on Poplar Veneer Surface and its Application in Plywood[J]. *Bioresources*. 2016, 11(1): 1571-1584.

[2] Chen W, Zhou X, Zhang X, et al. Fast enhancement on hydrophobicity of poplar wood surface using low-pressure dielectric barrier discharges (DBD) plasma[J]. *Applied Surface Science*. 2017, 407: 412-417.

[10] This paper pays close attention on the plasma modification, so only the C1s deconvoluted in Figure 6 is not enough, the N1 and O1 should also be done as well. one more thing, XPS analysis should have allowed a quantitative assignment of each kind of nitrogen/carbon/oxygen instead of just a qualitative approach.

Response: According to your helpful suggestion, we present the quantitative assignment of O1s and N1s as shown below (Fig. 3 and Table 1). As shown in Fig. 3(a-b), the O1s spectrum of CSAC and HCSAC can be resolved into three individual peaks based on the previous literature (531.3 eV, carbonyl and/or quinone; 532.7 eV, hydroxyl and/or ether; 534.0 eV, chemisorbed oxygen and/or water)^[1]. As shown in Fig. 3(c-d), the N1s spectrum can be fitted by three peaks (398.2 eV, pyridinic nitrogen; 399.9 eV, pyrrolic nitrogen; 400.9 eV, quaternary nitrogen)^[2]. The nitrogen content of HCSAC show slightly changes in comparison with CSAC. Table 1 presents the composition of the oxygen and nitrogen chemical functional groups of CSAC and HCSAC. The obvious increment of carbonyl and carboxyl can be ascribed to the ions excited from the H₂O molecule during H₂O plasma modification. The results from the deconvoluted C1s spectrum is more significant in this manuscript than that from O1s and N1s. However, according to your suggestion, we supply the results of deconvoluted O1s and N1s in supporting information.

Fig. 3 (a-b) High resolution XPS of O 1s peaks of CSCA and HCSAC. (c-d) High resolution XPS of N 1s peak of CSAC and HCSAC.

Table 1 Composition of oxygen and nitrogen chemical groups of CSAC and HCSAC.

		Assignment	CSAC	HCSAC
Components (%)				
O 1s				
O1	Carbonyl (-C=O) and/or quinone (marked as O-I)		27.6	32.5
O2	Hydroxyl (C-OH) and/or ether (C-O-C) (marked as O-II)		42.0	32.3
O3	Chemisorbed oxygen (COOH) and/or water (marked as O-III)		30.4	35.2
N 1s				
N1	Pyridinic nitrogen (marked as N-6)		25.6	27.6
N2	Pyrrolic nitrogen (marked as N-5)		38.0	35.7
N3	Quaternary nitrogen (marked as N-Q)		36.4	36.7

Reference:

- [1] Kai W, Ning Z, Lei S, et al. Promising biomass-based activated carbons derived from willow catkins for high performance supercapacitors[J]. *Electrochimica Acta*. 2015, 166: 1-11.
- [2] Long Q, Chen W, Xu H, et al. Synthesis of functionalized 3D hierarchical porous carbon for high-performance supercapacitor[J]. *Energy & Environmental Science*. 2013, 6(8): 2497-2504.

[11] Did the authors study the reusability for electrochemical properties? It is important to cost down the process.

Response: After taking your helpful suggestion into account, we supply the life cycles of HCSAC electrode at different current density for 1000 cycles, shown in Fig. 4. The capacitance retention of CSAC modified by H₂O plasma (HSCAC) electrodes maintain over 97% in the 6.0 M KOH electrolyte. The high capacitance retention indicates that ideal interconnected porous structure generating high ion transmission efficiency in HCSAC. Also, we supply this figure in supporting information.

Fig. 4 Life cycles of HCSAC electrode at different current density for 1000 cycles.

[12] It is necessary to compare the results with some recent literatures.

Response: According to your helpful suggestion, we provide the electrochemical properties (specific capacitance and rate capability (capacitance retention at the current density from 0.5 A g⁻¹ to 10 A g⁻¹)) of activated carbon in some recent literatures (shown in Table 2). The CSAC utilized in our experiment exhibits low specific capacitance can be ascribed to commercial activated carbon is generally physically activated by steam or carbon dioxide. The biomass derived carbon materials prepared from harsh experimental environment commonly adopt chemical activation (KOH activation adopted in the references below), listed in Table 2^[1-4]. The rate capability of CSAC was improved from 76.7% to 84.6% and an increment of 64.8% in specific capacitance after H₂O plasma modification. Furthermore, we are capable of applying this H₂O plasma modification in other activated carbon electrode materials to improve specific capacitance and rate capability.

Furthermore, the nitric acid treatment introducing oxygen containing functional group to activated carbon need long heating duration, extra energy consumption and utilization of chemical reagents^[5,6]. In comparison to the reported modifying method,

the H₂O plasma modification used in our experiment can improve the electrochemical properties of CSAC with a short time (100 s) and pollution-free media (H₂O).

Table 2 Comparison of electrochemical performance of activated carbon from biomass precursors.

Material	Specific capacitance	Rate capability	Electrolyte	Cell configuration	Reference
Broad bean shells	202 F g ⁻¹ at 0.5 A g ⁻¹	63.9% (from 0.5 A g ⁻¹ to 10 A g ⁻¹)	6 M KOH	3-electrode	[1]
Human hair	228 F g ⁻¹ at 0.25 A g ⁻¹	73.0% (from 0.5 A g ⁻¹ to 6 A g ⁻¹)	6 M KOH	3-electrode	[2]
Pectin	231 F g ⁻¹ at 1 A g ⁻¹	63.0% (from 1 A g ⁻¹ to 10 A g ⁻¹)	6 M KOH	3-electrode	[3]
Pomelo peel	342 F g ⁻¹ at 0.2 A g ⁻¹	62.0% (from 0.2 A g ⁻¹ to 20 A g ⁻¹)	6 M KOH	3-electrode	[4]
HCSAC	155.2 F g ⁻¹ at 1 A g ⁻¹	84.6% (from 0.5 A g ⁻¹ to 10 A g ⁻¹)	6 M KOH	3-electrode	This work

Reference:

- [1] Guiyin X, Jinpeng H, Bing D, et al. Biomass-derived porous carbon materials with sulfur and nitrogen dual-doping for energy storage[J]. *Green Chemistry*. 2015, 17(3): 1668-1674.
- [2] Si W, Zhou J, Zhang S, et al. Tunable N-doped or dual N, S-doped activated hydrothermal carbons derived from human hair and glucose for supercapacitor applications[J]. *Electrochimica Acta*. 2013, 107: 397-405.
- [3] Fan Y, Liu P F, Yang Z J, et al. Bi-functional porous carbon spheres derived from pectin as electrode material for supercapacitors and support material for Pt nanowires towards electrocatalytic methanol and ethanol oxidation[J]. *Electrochimica Acta*. 2015, 163: 140-148.
- [4] Qinghua L, Ling Y, Zheng-Hong H, et al. A honeycomb-like porous carbon derived from pomelo peel for use in high-performance supercapacitors[J]. *Nanoscale*. 2014, 6(22): 13831-13837.
- [5] Xie Y B, Qiao W M, Zhang W Y, et al. Effect of the surface chemistry of activated carbon on its electrochemical properties in electric double layer capacitors[J]. *New Carbon Materials*. 2010, 25(4): 248-254.
- [6] Nian Y R, Teng H S. Nitric acid modification of activated carbon electrodes for improvement of electrochemical capacitance[J]. *Journal of The Electrochemical Society*. 2002, 149(8): A1008-A1014.

[13] Deeper discussion of the results should be provided.

Response: Based on your significant suggestion, we supply the three-dimensional schematic model of the functional groups of HCSAC (shown in Fig. 5). The oxygen functional groups and nitrogen functional groups are presented in the figure below.

The additional pseudo-capacitance is attributed to oxygen functional groups. Previous studies demonstrate that some oxygen functional groups can directly participate in faradaic reactions not only in acidic medium^[1] but alkaline medium^[2]. The increased oxygen functional groups in HCSAC play an important role in the enhancement of capacitance via reversible redox actions in 6 M KOH electrolyte. The inductive effects of oxygen containing functional groups' bonds structure is capable of causing the electrons redistribution and some bonds polarization. The electric potential induced

redox reactions of polarized sites proceed through the simultaneously reversible gaining/losing of electrons and adsorption/desorption of protons, respectively^[3,4]. This deeper discussion has been added to the ‘Result and discussion’ section. The supporting references are listed below.

Reference:

- [1] Hsieh C T, Teng H. Influence of oxygen treatment on electric double-layer capacitance of activated carbon fabrics[J]. *Carbon*. 2002, 40(5): 667-674.
- [2] Tang C, Liu Y, Yang D, et al. Oxygen and nitrogen co-doped porous carbons with finely-layered schistose structure for high-rate-performance supercapacitors[J]. *Carbon*. 2017, 122: 538-546.
- [3] Chen C M, Zhang Q, Zhao X C, et al. Hierarchically aminated graphene honeycombs for electrochemical capacitive energy storage[J]. *Journal of Materials Chemistry*. 2012, 22(28): 14076-14084.
- [4] Liu B, Liu Y, Chen H, et al. Oxygen and nitrogen co-doped porous carbon nanosheets derived from *Perilla frutescens* for high volumetric performance supercapacitors[J]. *Journal of Power Sources*. 2017, 341: 309-317.

Fig. 5 Three-dimensional schematic model of the functional groups of HCSAC.

Reviewer #2

[1] Typos like missing space between value and unit (70 °C, line 46, P2, 5.0mm, line 47, P2 etc.), capital letter (R_{ct}, line 46, P4) are detected. Check carefully!

Response: According to your helpful suggestions, we carefully check the space between value, unit and capital letter through the whole paper and correct them. We have added the space between value and unit. The sentence of ‘... 70 °C ...’ has been revised to ‘...70 °C...’. Similar typos are modified in our manuscript. Meanwhile, the ‘R_{ct}’ and ‘R_i’ have been revised to ‘R_{ct}’ and ‘R_i’, respectively.

Page 2, line 52 --- ‘A water bath was used to heat H₂O to 70 °C and a vacuum pump was applied in keeping the pressure of the reactor chamber to 30 KPa’.

Page 4, line 60 and page 5, line 3 --- ‘The intercept on real axis is related to the internal resistance (R_i)...’ and ‘The R_i of samples...’.

Page 5, line 7 and line 8 --- ‘With the frequency decreasing, the semicircle represents the charge transfer resistance (R_{ct})...’ and ‘The smaller diameter in the semicircle of HCSAC electrodes illustrated the lower R_{ct}...’.

[2] Abbreviations should be denoted at their first appearance in main text, e.g. CSAC. EDLC is commonly the abbreviation of “electrical double layer capacitor” rather than electrical double layer capacitance.

Response: After taking your helpful suggestion into account, we carefully check the abbreviations. The ‘DBD’ and ‘HCSAC’ were revised to ‘dielectric barrier discharge (DBD)’ and ‘CSAC modified by H₂O plasma (HCSAC)’ when they first appear in main text, respectively. We check the meaning of ‘EDLC’ based on the previous literature. Afterwards, ‘electrical double layer capacitance (EDLC)’ has been revised to ‘electrical double layer capacitor (EDLC)’.

Page 2, line 13 --- ‘while dielectric barrier discharge (DBD) can be operated at atmospheric pressure’.

Page 3, line 12 --- ‘see XPS spectra of CSAC and CSAC modified by H₂O plasma (HCSAC) in Fig. S1’.

[3] The Fig. numbers are in mess, do not match with the description in text.

Response: Based on your helpful suggestion, we carefully check the valid numerals of our tables and figures through the whole paper and corrected them. The related number

of figures and tables through the whole paper have been revised accordingly as presented below.

Page 3, line 41 --- 'Figure 2 shows N₂ adsorption/desorption isotherms and the DFT pore size distributions of CSAC and HCSAC.'

Page 3, line 57 --- 'Figure 3 demonstrates two obvious bands corresponding to the G-band (1530-1610 cm⁻¹) and D-band (1320-1370 cm⁻¹) of CSAC and HCSAC.'

Page 4, line 2 --- 'Morphology of CSAC and HCSAC were further studied by TEM, shown in Figure 4.'

Page 4, line 7 --- 'FTIR was used in this study (shown in Figure 5) to investigate the surface chemistry properties of SCAC and HCSAC.'

Page 4, line15 --- 'The surface elemental compositions of CSAC and HCSAC were further investigated by X-ray photoelectron spectroscopy (XPS), shown in Figure 6.'

Page 4, line 34 --- 'Fig. 7a shows the CV plots of the CSAC and HCSAC electrodes at a sweep rate of 10mV s⁻¹.'

Page 4, line 51 --- 'The GCD curves (Fig. 7b) of CSAC and HCSAC at a current density of 1A g⁻¹ show typical triangular shapes.'

Page 4, line 56 --- 'To further comprehend the capacitive behavior of SCAC and HSCAC, EIS test was performed over a frequency range from 10 kHz to 10 mHz (Fig. 7c).'

[4] Fig. 7d (Fig. 6 in manuscript) is not mentioned. Looks like it is the capacitance performance test at different current density, however, more cycles, e.g. 1000 cycles for each current density, should be provided. Otherwise, it is not sufficient to prove the good performance at different current densities.

Response: According to your kind remind, we realize that Fig. 7d (Fig. 6 in manuscript) has been analyzed, but we forget to mention this figure number. we add the figure number 'As shown in Fig. 7d' before explanation.

After considering your significant suggestions, we add the life cycles of HCSAC electrode at different current density (from 0.5 A g⁻¹ to 10 A g⁻¹) for 1000 cycles, shown in Fig. 1. The capacitance retention of CSAC modified by H₂O plasma (HSCAC) electrode maintains over 97% in the 6.0 M KOH electrolyte. The high capacitance retention indicates that ideal interconnected porous structure generating high ion transmission efficiency in HCSAC. Also, we supply this figure in supporting information.

Fig. 1 Life cycles of HCSAC electrode at different current density for 1000 cycles.

[5] In electrochemical properties section, the explanation of improved capacitance performance is concerned. It is claimed by authors that the improvement is attributed to pseudo-capacitance owing of oxygen functional groups on carbon surface (line 22-23, P4) besides double layer capacitance. How did the pseudo capacitance of oxygen functional groups work? What is the possible working mechanism? And any supporting references?

Response: Based on your significant suggestions, we provide the possible working mechanism of oxygen functional groups in the manuscript. The additional pseudo-capacitance is attributed to oxygen functional groups. Previous studies demonstrate that some oxygen functional groups can participate directly in faradaic reactions not only in acidic medium^[1] but alkaline medium^[2]. The increased oxygen functional groups in HCSAC play an important role in the enhancement of capacitance via reversible redox actions in 6 M KOH electrode. The inductive effects of oxygen containing functional groups' bonds structure is capable of causing the electrons redistribution and some bonds polarization. In 6 M KOH electrolyte, the electric potential induced redox reactions of polarized sites proceed through the simultaneously reversible gaining/losing of electrons and adsorption/desorption of protons, respectively^[3,4]. This deeper discussion has been added to the 'Result and discussion' section. The supporting references are listed below.

Page 4, line 38 --- 'The additional pseudo-capacitance is attributed to oxygen functional groups. Previous studies demonstrate that some oxygen functional groups can directly participate in faradaic reactions not only in acidic medium but alkaline medium. The increased oxygen functional groups play an important role in the enhancement of

capacitance via reversible redox actions in alkaline medium. The inductive effects of oxygen containing functional groups' bonds structure is capable of causing the electrons redistribution and some bonds polarization. In 6 M KOH electrolyte, the electric potential induced redox reactions of polarized sites proceed through the simultaneous reversible gaining/losing of electrons and adsorption/desorption of protons, respectively'.

Reference:

- [1] Hsieh C T, Teng H. Influence of oxygen treatment on electric double-layer capacitance of activated carbon fabrics[J]. Carbon. 2002, 40(5): 667-674.
- [2] Tang C, Liu Y, Yang D, et al. Oxygen and nitrogen co-doped porous carbons with finely-layered schistose structure for high-rate-performance supercapacitors[J]. carbon. 2017, 122: 538-546.
- [3] Chen C M, Zhang Q, Zhao X C, et al. Hierarchically aminated graphene honeycombs for electrochemical capacitive energy storage[J]. Journal of Materials Chemistry. 2012, 22(28): 14076-14084.
- [4] Liu B, Liu Y, Chen H, et al. Oxygen and nitrogen co-doped porous carbon nanosheets derived from *Perilla frutescens* for high volumetric performance supercapacitors[J]. Journal of Power Sources. 2017, 341: 309-317.

[6] Contact angel should be measured to support the claimed improved hydrophilicity and wettability after plasma treatment;

Response: Based on your helpful suggestions, we try to carry on the contact angle measurement. We have failed in pressing the powder into full tablet and thus the test results are not convincing. However, we have found supporting references to support the claim that hydrophilicity and wettability are improved after plasma modification. The oxygen functional groups can provide an extra pseudo-capacitance through revisable redox reaction and improved wettability between the electrodes and electrolytes^[1-3]. The presence of surface oxygen functional groups is advantageous for improving the hydrophilicity of carbon material which would increase the surface area accessible to aqueous electrolyte^[4-6].

Reference:

- [1] Zhao Y Q, Lu M, Tao P Y, et al. Hierarchically porous and heteroatom doped carbon derived from tobacco rods for supercapacitors[J]. Journal of Power Sources. 2016, 307: 391-400.
- [2] Si W, Zhou J, Zhang S, et al. Tunable N-doped or dual N, S-doped activated hydrothermal carbons derived from human hair and glucose for supercapacitor applications[J]. Electrochimica Acta. 2013, 107: 397-405.
- [3] Long Q, Chen W, Xu H, et al. Synthesis of functionalized 3D hierarchical porous carbon for high-performance supercapacitor[J]. Energy & Environmental Science. 2013, 6(8): 2497-2504.
- [4] Fan Y, Liu P F, Yang Z J, et al. Bi-functional porous carbon spheres derived from pectin as electrode material for supercapacitors and support material for Pt nanowires towards electrocatalytic methanol and ethanol oxidation[J]. Electrochimica Acta. 2015, 163: 140-148.
- [5] Kai W, Ning Z, Lei S, et al. Promising biomass-based activated carbons derived from willow catkins for high performance supercapacitors[J]. Electrochimica Acta. 2015, 166: 1-11.

[6] Chang J, Gao Z, Wang X, et al. Activated porous carbon prepared from paulownia flower for high performance supercapacitor electrodes[J]. *Electrochimica Acta*. 2015, 157: 290-298.

[7] How is the plasma working condition come up, e.g. 160 W, 100 s, 30 KPa? How did these factors affect the treatment and consequently to capacitance performance? In my experience, the treatment time is highly related to the structure evolution.

Response: According to your helpful suggestions, we have provided more electrochemical performance data of samples prepared under different time (50 s, 100 s, 150 s, 200 s, 300 s) and power (50 W, 100 W, 130 W, 160 W, 200 W), please see the figure below. We have conducted a series of preliminary experiment on electrochemical performance to select the optimum condition (100 s, 160 W). From the results shown below, it can be concluded that the H₂O dielectric barrier discharge (DBD) plasma has ability to improve electrochemical performance of commercial coconut shell-based activated carbon (CSAC) in a short time. The exhaustive study is very meaningful that we are going to investigate in the future.

The electrochemical performance of CSAC modified with H₂O plasma under different time and different power is shown below (Fig. 2). The cyclic voltammetry (CV) plots at the scan rate of 10 mV s⁻¹ of CSAC and CSAC modified with H₂O plasma are shown in Fig. 2(a-b). The integrated area of the sample D-100 W-100 s (D means DBD plasma modification, the first number is modification power, the second number is modification time) is larger than other sample in the same power, indicating that 100 s is a more suitable modification time. With modification power changing from 50 W to 200 W, the sample D-160 W-100 s exhibits the largest integrate area among all the samples in the same modification time of 100 s. The largest integrate area of D-160 W-100 s indicates that the sample shows the larger specific capacitance than other samples. Furthermore, the galvanostatic charge-discharge (GCD) curves of CSAC and CSAC modified with H₂O plasma at the current density of 1 A g⁻¹ are shown in Fig. 2(c-d). After calculation by the formula below, we come up the conclusion that D-160 W-100 s exhibits the excellent specific capacitance among all the samples, which is consistent with CV curves. The Nyquist plots of CSAC and CSAC modified with different power exhibit similar shapes, shown in Fig. 2e. The internal resistance (R_i) of samples were below 0.4 Ω , demonstrating a good electrical conductivity of materials. The increased R_i of the sample D-160 W-100 s can be ascribed to the presence of surface oxides and thus increase the ohmic resistance along the axial direction of micropores. Also, the smaller diameter in the semicircle of the sample D-160 W-100 s illustrates that the lower charge transfer resistance (R_{ct}) at electrolyte interfaces, owing to the higher hydrophilicity after oxygen functional groups introduced. Fig. 2f demonstrates the rate capability (capacitance retention from the current density of 0.5 A g⁻¹ to 10 A g⁻¹) of CSAC and CSAC modified under different power. The higher rate capability indicates that the sample is capable of maintaining excellent charge and discharge characteristics at high current density. In summary, we perform other detailed characterization of

CSAC modified by H₂O plasma under modification power of 160 W and time of 100 s (HCSAC) based on the electrochemical performance. We keep the pressure of the reactor chamber to 30 KPa, mainly aiming to pump water into the reactor chamber instead of maintaining the vacuum environment. Based on the above results, we select the conditions (discharge power, 160 W; treating time, 100 s; the pressure, 30 KPa) to make detailed analysis. We supply the result in supporting information.

The specific capacitance of the electrodes (C), was calculated by the following equation:

$$C = \frac{I \times \Delta t}{\Delta V \times m}$$

Where C is specific capacitance (F g⁻¹), I is the constant charge-discharge current (A), Δt is the discharge time (s), ΔV is the total change in voltage (V), m is the mass of the active material in an electrode (g)

Fig. 2 (a-b) CV curves of CSAC and CSAC modified with H₂O plasma in a three- system with 6 M KOH aqueous electrolyte at the scan rate of 10 mV s⁻¹. (c-d) GCD curves of CSAC and CSAC modified with H₂O plasma in a three-electrode system with 6 M KOH aqueous electrolyte at the current density of 1 A g⁻¹. (e) Nyquist plots of CSAC and CSAC modified at different power electrodes. The inset is the detail with enlarged scale (f) Rate capability of CSAC and HCSAC modified at different power at the current density from 0.5 to 10 A g⁻¹.

[8] In the XPS analysis result (Table 2), why is the atomic composition of N increasing?

Response: The definite parameters of our experiment are determined by a series of preliminary experiments. The results shown in this article were obtained by one experiment. According to your helpful suggestion, we consider that the increasing nitrogen content may be ascribed to experimental error. We perform X-ray photoelectron spectroscopy (XPS) test to CSAC and HCSAC for five times,

respectively. Afterwards, we delete smallest and largest values and take the average of the three times results. The nitrogen atomic composition is listed below (Table 1), the average value of CSAC and HCSAC is 1.04% and 1.58%, respectively (the value in our manuscript of CSAC and HCSAC is 1.6% and 2.8%, respectively). The nitrogen content is much lower when compared with oxygen and carbon content. So, nitrogen play a relatively minor role in this experiment.

Table 1 The nitrogen atomic composition of CSAC and HCSAC

Sample	Nitrogen atomic composition (%)
	0.55
	1.11
CSAC	0.96
	1.06
	1.13
Average	1.04
	0.66
	0.96
HCSAC	2.19
	1.59
	3.02
Average	1.58

Appendix B

Responses to reviewer's comments

We appreciate the detailed and helpful comments and suggestions which are very helpful to improve the quality of our manuscript. The manuscript (RSOS-180872.R2) has been carefully revised as required. All new and revised contents are highlighted with yellow in our revised version. We hope that the correction will meet with your approval. The point-by-point answers to the comments and suggestion were listed below.

Reviewer #2

[1] I am satisfied with the modifications made by authors according to my comments, and it can be accepted on condition the supplementary figures, discussions and references are wrapped up with the manuscript. Otherwise, the manuscript is quite short, hardly to be a full-length research paper with few substantial contents.

Response: According to your helpful suggestion, we supplied the part of supplementary figures, discussions and references into our revised manuscript. For example, the quantitative assignment of O1s and N 1s, the three-dimensional schematic model of the functional groups of HCSAC (commercial coconut shell-based activated carbon modified with H₂O plasma) and the life cycles of HCSAC electrode at different current density for 1000 cycles were supplied in the revised manuscript, shown in Fig.1, Fig. 2, Fig. 3 and Table below. (Listed as Fig. 6, Fig. 7, Fig. 9 and Table 2 in our revised manuscript, respectively.).

Page 4, line 17 --- ‘As shown in Fig. 6(c-d), the O 1s spectrum of CSAC and HCSAC can be resolved into three individual peaks based on the previous literature...’.

Page 4, line 27 --- ‘The three-dimensional schematic model of oxygen functional groups and nitrogen functional groups of HCSAC are presented in Fig. 7’.

Page 5, line 14 --- ‘The life cycles of HCSAC at different current density for 1000 cycles to prove the reusability of HCSAC can be seen in Fig. 9. The capacitance retention of CSAC...’.

Fig. 1 (a-b) High resolution XPS of O 1s peaks of CSCA and HCSAC. (c-d) High resolution XPS of N 1s peak of CSAC and HCSAC.

Table 1 Composition of oxygen and nitrogen chemical groups of CSAC and HCSAC.

		Assignment	CSAC	HCSAC
Components (%)				
O 1s				
O1	Carbonyl (-C=O) and/or quinone (marked as O-I)		27.6	32.5
O2	Hydroxyl (C-OH) and/or ether (C-O-C) (marked as O-II)		42.0	32.3
O3	Chemisorbed oxygen (COOH) and/or water (marked as O-III)		30.4	35.2
N 1s				
N1	Pyridinic nitrogen (marked as N-6)		25.6	27.6
N2	Pyrrolic nitrogen (marked as N-5)		38.0	35.7
N3	Quaternary nitrogen (marked as N-Q)		36.4	36.7

Fig. 2 Three-dimensional schematic model of the functional groups of HCSAC.

Fig. 3 Life cycles of HCSAC electrode at different current density for 1000 cycles.

[2] Did the authors measure the mass of samples before and after plasma treatment? Besides, 160 W is not low and there will be mass loss based on my experience on plasma processing.

Response: Based on your significant suggestion, we re-weighed the mass of samples before and after treatment.

The weight loss rate of sample (W), was calculated by the following equation:

$$W = \frac{x - y}{z}$$

Where x is the weight of the whole reactor chamber including the sample before plasma treatment (g), y is the weight of the whole reactor chamber including the sample after plasma treatment (g), z is the weight of the sample.

After weighing for three times, the average of x was 138.4310 g, the average of y was 138.4309 g, the weight of sample is 0.5 g, the difference between x and y is too small and beyond their accuracy. The results indicate that 160 W can hardly cause weight loss of activated carbon.